# Merging Decision Transformers: Weight Averaging for Forming Multi-Task Policies

**Daniel Lawson & Ahmed H. Qureshi**
Department of Computer Science
Purdue University
West Lafayette, IN 47907, USA
`{lawson95; ahqureshi}@purdue.edu`

## Abstract

Recent work has shown the promise of creating generalist, transformer-based, policies for language, vision, and sequential decision-making problems. To create such models, we generally require centralized training objectives, data, and compute. It is of interest if we can more flexibly create generalist policies, by merging together multiple, task-specific, individually trained policies. In this work, we take a preliminary step in this direction through merging, or averaging, subsets of Decision Transformers in weight space trained on different MuJoCo locomotion problems, forming multi-task models without centralized training. We also propose that when merging policies, we can obtain better results if all policies start from common, pre-trained initializations, while also co-training on shared auxiliary tasks during problem-specific finetuning. In general, we believe research in this direction can help democratize and distribute the process of which forms generally capable agents.

## 1 Introduction

Transformers, specifically those pre-trained with language, have been shown to learn general representations and parameters that are amenable to transfer (Lu et al., 2021). In the context of offline Reinforcement Learning, Reid et al. (2022); Takagi (2022) showed that initializing Decision Transformers (DT) (Chen et al., 2021; Furuta et al., 2021) with pre-trained language models can increase convergence speed and performance on the D4RL (Fu et al., 2020) MuJoCo (Todorov et al., 2012) benchmarks, showing transfer between completely different modalities. Transformers can also learn generalist policies for different Atari games in Multi-game Decision Transformer (Lee et al., 2022), as well as for many modalities, tasks, and embodiments in Gato (Reed et al., 2022). Keeping our focus on Decision Transformers, we question the nature of which multi-game, or more generally, multi-task Decision Transformers are formed. Is it possible to more flexibly create multi-task DTs with reduced demands for centralizing all data and training?

We believe a small step in this direction is investigating parameter similarities between single Decision Transformers trained on different reinforcement learning problems. Specifically, In this work, we look at DTs trained on different MuJoCo locomotion problems. We begin by analyzing the similarity of learned parameters but through a perspective of weight merging. Specifically, given two trained DTs on different environments, merging refers to taking a subset of parameters, say those associated with attention, and replacing the parameters in both models with an average of original parameters. In this work, we look at merging without accounting for symmetries (Ainsworth et al., 2022), but this could be a future direction.

We aim to answer several questions, forming our contributions:

1. We begin by investigating how merging individual layers and subsets of DTs affect model performance and find that we can directly merge and swap the parameters of Decision Transformers trained on different MuJoCo environments (HalfCheetah, Walker2D, Hopper) with, in some cases, minimal decrease in performance. This leads us to investigate the role of attention in DTs, finding some DTs do not heavily rely on attention.

2. We propose a method for creating multi-task DTs through merging certain parameters, freezing those merged, and then independently finetuning un-merged parts (*Merge-Freeze-Finetune*). This creates a multi-task DT without centralized data or training objectives.

3. We propose that common initialization and shared co-training on auxiliary tasks can lead to better merging with Decision Transformers. Specifically, we use language modeling, building on Reid et al. (2022).

## 2 BACKGROUND

### 2.1 TRANSFORMERS

Transformers (Vaswani et al., 2017) are a common neural network architecture for modeling sequences. We consider causal decoder-only transformers like GPT (Radford & Narasimhan, 2018). A transformer is composed of several successive transformer blocks, which each consist of (multi-headed) self-attention layers, multi-layer perceptron (MLP) layers, as well as layer normalization (Ba et al., 2016). Given a sequence of inputs of length $n$ and embedding size $d$ $X \in \mathbb{R}^{d \times n}$, a self-attention layer projects each input to queries ($Q$), keys ($K$), and values ($V$) with parameters $\{W_q, b_q\}, \{W_k, b_k\}, \{W_v, b_v\}$ respectively. We then perform Attention($X$): softmax($\frac{QK^T}{\sqrt{d}}$)$V$. We apply this operation for each head in parallel, stack outputs, resulting in $Y \in \mathbb{R}^{Hd \times n}$, where $H$ is the number of heads, which is then projected by $W_o$, resulting in $X' = W_o Y \in \mathbb{R}^{d \times n}$ (Phuong & Hutter, 2022). Decision Transformers, apply layer normalization before and after attention layers, where each layer has learnable affine transform parameters $\gamma, \beta$. Each MLP contains one hidden layer (and one output layer), with parameters $W_1, W_2$. Transformers also have a residual connection after each attention and MLP layer respectively, so the actual output consists of the transformation from the layer added with its input.

### 2.2 OFFLINE REINFORCEMENT LEARNING WITH DECISION TRANSFORMER

We consider problems that are modeled by a Markov decision process (MDP) with states $s \in \mathcal{S}$, actions $a \in \mathcal{A}$, unknown transition dynamics $p(s'|s, a)$ and reward function $r(s, a)$. Traditionally in RL, we aim to learn an optimal policy that maximizes expected (discounted) return through interaction with the environment. Instead, in offline RL, we learn without interaction using a static dataset. A dataset consists of a set of trajectories, where each trajectory has the form $\tau = (s_0, a_0, r_0, s_1, a_1, r_1, ..., s_N, a_N, r_N)$, consisting of a sequence of states, actions, and return at each timestep until timestep $N$. With this data, we aim to find a policy $\pi(a|s, \cdot)$ that maximizes expected return $\mathbb{E}[\sum_{t=0}^N r_t]$, where the expectation is over the distribution induced by transition dynamics and the policy $\pi$. Decision Transformer casts offline RL as a sequence modeling problem using causal auto-regressive transformers. Particularly, DT models the actions in the sequence $\tau = (\hat{R}_1, s_1, a_1, \hat{R}_2, s_2, a_2, \ldots, \hat{R}_T, s_T, a_T)$, where the return-to-go (RTG) is the undiscounted sum of future reward: $\hat{R}_t = \sum_{t'=t}^T r_{t'}$. To make a prediction, each input is embedded using linear projections with added positional encoding and normalization, passed through the transformer, and then a final linear layer predicts the action. We refer to transformer parameters as those only associated with transformer layers and not input/output projections. During test time, we condition on an initial RTG such as the maximum in the dataset multiplied by some constant, querying the DT to generate an action that leads to high-quality performance. At each timestep, we condition on a sequence consisting of (truncated) past transitions, as well as the remaining RTG to achieve.

### 2.3 MERGING

Given two trained neural networks with the same architecture, and their sets of parameters or weights $\Theta_A, \Theta_B$, we can interpolate between weights, getting a new model $\Theta_\lambda = (1 - \lambda)\Theta_A + \lambda\Theta_B$. With $\lambda = .5$, we directly average weights, obtaining a merged model $\Theta_{\lambda=.5} = \Theta_M = .5(\Theta_A + \Theta_B)$. Directly interpolating between parameters of randomly initialized models results in models with high loss (Entezari et al., 2021; Ainsworth et al., 2022). Instead of applying merging over all parameters, we can apply this same principle over a subset of parameters. In this work, we are interested in merging transformer parameters discussed in Section 2.1.

In the case that models $\Theta_A, \Theta_B$ are from two finetuned models from a common initialization, $\Theta_{\text{pre}}$, merging is equivalent to task arithmetic for creating multi-task models (Ilharco et al., 2022). Specifically, a task vector $\tau_x = \Theta_x - \Theta_{\text{pre}}$ stores the difference between a finetuned and pre-trained model. We can form a multi-task model by adding the averaged task vectors to the original model: $\Theta_{\text{pre}} + .5(\tau_A + \tau_B) = \Theta_{\text{pre}} + .5((\Theta_A - \Theta_{\text{pre}}) + (\Theta_B - \Theta_{\text{pre}})) = .5(\Theta_A + \Theta_B) = \Theta_M$. While discussed prior work considers merging models trained on the same or overlapping data distributions, we begin by using merging as a tool to see the similarity of DTs trained on different data.

## 3 Merging in Offline RL with Decision Transformers

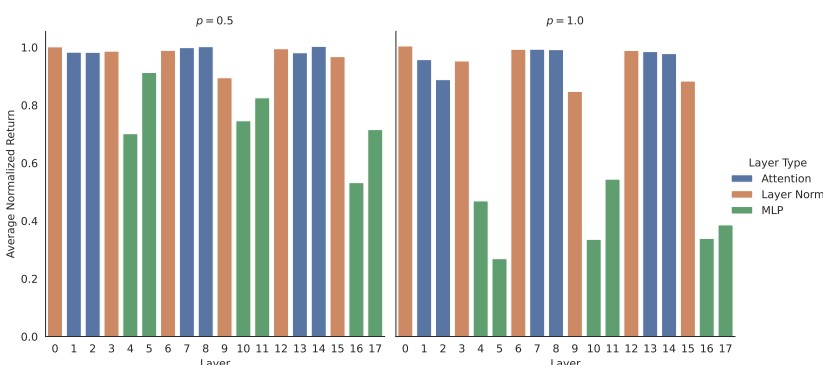

Figure 1: Normalized return after **merging a single layer** at a time, averaged between all pairs of Walker, Hopper, HalfCheetah. Return is normalized by original performance before merging. We evaluate at $p = .5$. an average, and $p = 1$, swapping in a layer from another DT.

We begin by partially merging Decision Transformers to investigate if similar parameters may be learned across environments, ignoring input and output projection layers which have unique dimensionality and function for different environments as each robot has unique state and action spaces. We use settings for MuJoCo experiments from Chen et al. (2021); Reid et al. (2022), as well as their implementations. Following their configuration, we use transformers with an embedding size of 128, 1 head, and 3 layers. We provide further hyperparameters in Appendix A. We randomly initialize and train a model for HalfCheetah, Hopper, and Walker2D on Expert D4RL datasets. Given two trained models on different environments, we call the model which is having a layer altered the target model with parameters $\Theta_t$, and the model where the layer is taken from, which is trained on another environment, the source model with parameters $\Theta_s$.

When averaging a single layer at a time, we keep all other layers in the target model unaltered, and then rollout the policy (for 25 episodes), evaluating the impact of partially, or fully, utilizing the parameters from the other model. For one layer $\theta^i \in \Theta$, we use the update $\theta_t^i = (1 - p)\theta_t^i + p(\theta_s^i)$, where $p \in \{.5, 1\}$, relating to averaging and swapping. We merge between each pair of models, and for each pair, we look at merging in each direction, swapping source and target models. This leads to six instances of evaluation per layer, which we average and display in Figure 1. We report normalized return, where 1 corresponds to the original return before any merging ($p = 0$).

We can both average ($p = .5$), or directly use a layer from another transformer ($p = 1$), showing that Decision Transformers trained on different MuJoCo tasks may learn functionally similar parameters. We see a larger drop-off from merging parameters within the multi-layer perception (MLP) layers of the transformer as opposed to attention parameters. Additionally, as we go further in depth or set $p = 1$, we see a drop-off when merging MLPs. Surprisingly, we can directly use the attention weights from another trained model at any depth with little to no decrease in empirical return.

### 3.1 Similarity in Parameter Space

Over the same experiments in Figure 1, we report the $L_2$ distance between the source and target model, shown in Figure 2. We see that $L_2$ distance does not quite correlate with better results. For

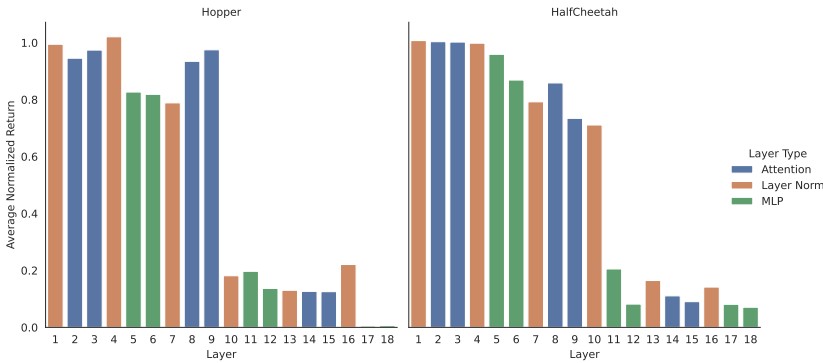

Figure 3: **Incremental merging** with $p = .5$ merging from HalfCheetah to Hopper (**left**) and Hopper to HalfCheetah (**right**), where the set of merged layers grows as moving further in depth.

example, the first attention layer consisting of weights of key, query, and value projections, has a large distance, comparable to that of the MLP layers. However, when merging this layer, we see a minimal decrease in return, while we saw reduced return when merging MLP layers.

## 3.2 MERGING SUBSETS OF MULTIPLE LAYERS

We should draw attention to that in Figure 1 we are only merging one layer at a time, and possibly, a transformer could be robust to single-layer interference. We must see how well merging performs with multiple layers at a time, as errors could compound. We look at this from two perspectives. We begin by following a similar procedure to the former per-layer merging, but instead incrementally add an additional layer, forming a larger subset as we go move further in depth, in Figure 3. Instead of averaging across different environment pairs, we show this within a single pair to be able to clearly see the trajectory formed by adding layers. We see that as we add layers, performance decreases, reaching close to zero performance when merging the entire transformer. We also tend to see large drop-offs at MLP layers, but less so at attention layers, and varying results with layer normalization.

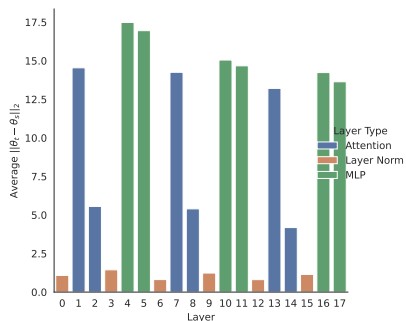

Figure 2: $L_2$ **distance between source and target models**, averaged over bidirectional pairs of HalfCheetah, Hopper, and Walker2D.

### 3.2.1 MERGING ALL ATTENTION LAYERS

When merging individual layers, we see the least reduction in performance with attention layers. Thus, we see what happens when merging all attention layers (the parameters associated with the query, key, value, and output linear projections). Attention parameters consist of $\sim 33.18\%$ of the 3 transformer layers which have a total size of 596K parameters. We display results in Figure 4. On the x-axis, we interpolate between parameters where $\lambda = 0, 1$ correspond to using attention parameters from just one environment. We hold unmerged parameters constant and evaluate on both environments at each $\lambda$. It appears that Decision Transformers trained on different MuJoCo can learn functionally similar attention weights. We can directly swap in the weights of HalfCheetah into the Walker2D transformer, and vice versa, with no decrease in return. We do not see this hold as strongly between Hopper/Walker2d, and Hopper/HalfCheetah, but we still see good results.

## 3.3 MERGE, FREEZE, FINETUNE

In previous sections, even when merging over a compatible subset, such as attention layers, we still see some drop-off in return after merging (such as between Hopper and HalfCheetah), as shown

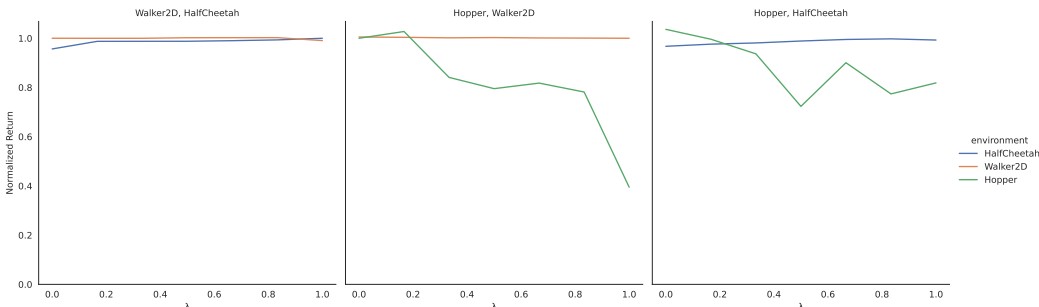

Figure 4: **Merging all attention parameters** of transformer across pairs of MuJoCo environments. $\lambda$ is varied, which interpolates between just parameters for the first environment $\lambda = 0$ and just parameters from the second environment $\lambda = 1$. Other parameters are unique for each environment.

in Figure 4. We aim to see if we can merge without losing performance, proposing *Merge-Freeze-Finetune* (MFF): Merging over a subset of two networks, freezing these layers, and then separately finetuning transformer un-merged parameters and linear input/output projections in both original models, to separately adapt to changes in shared parameters. We test this idea on several subsets, reporting the performance of the merged model as a percent of the original performance. We also report the size increase of the multi-task model within transformer layers, which does not include linear input/output projections that are kept unique for each task. For example, without any merging, we maintain original performance ($100\%$ on each task) but need unique transformer layers for each task ($200\%$). We use subsets of all attention related parameters, MLP and attention parameters, and the entire transformer which also adds layer normalization layers. We also report attention merging without finetuning (M), as previously shown in Section 3.2.1, and an alternative to equally merging, where we keep one transformer unmodified, but copy and freeze its parameters to the DT in the second environment, and have this second DT finetune its non-transformer parameters.

| Configuration | Hopper | HalfCheetah | Transformer Size |
|---|---|---|---|
| Original | 100% | 100% | 200% |
| Frozen Transformer from Hopper | 100% | 3% | 100% |
| Frozen Transformer from HalfCheetah | 48% | 100% | 100% |
| Transformer (MFF) | 61% | 86% | 100% |
| Attention+MLP (MFF) | 90% | 98% | 100.26% |
| Attention (MFF) | 101% | 101% | 166.82% |
| Attention (M) | 79% | 98% | 166.82% |

Table 1: Evaluating performance after merging between HalfCheetah and Hopper Expert models with *Merge-Freeze-Finetune* (MFF) and just merging (M), over different model subsets.

When merging with attention plus finetuning (MFF), we can recover the original performance. We can also retain greater compression with Attention+MLP merging, obtaining a multi-task model, with only 1.0026 times more transformer parameters, but with some reduced performance. Later, we see how we can improve with common initialization and co-training in Section 3.5.

## 3.4 ANALYSING ATTENTION MERGING

We aim to understand why attention merging is successful. Particularly, we find it surprising that we can swap the parameters of a DT trained on one environment with another and have a relatively small or no decrease in performance. To understand this phenomenon, we begin by visualizing attention weight patterns. This refers to the weights produced after softmax, which specify how much each input attends to other inputs. Because DT uses a causal mask, each input in a sequence can only attend to itself and previous inputs. We look at merging between Walker2d and HalfCheetah, using the same models previously used in Figure 4. We visualize using a context size consisting of 10 transitions (where each transition consists of RTG, states, and actions). We plot attention weights

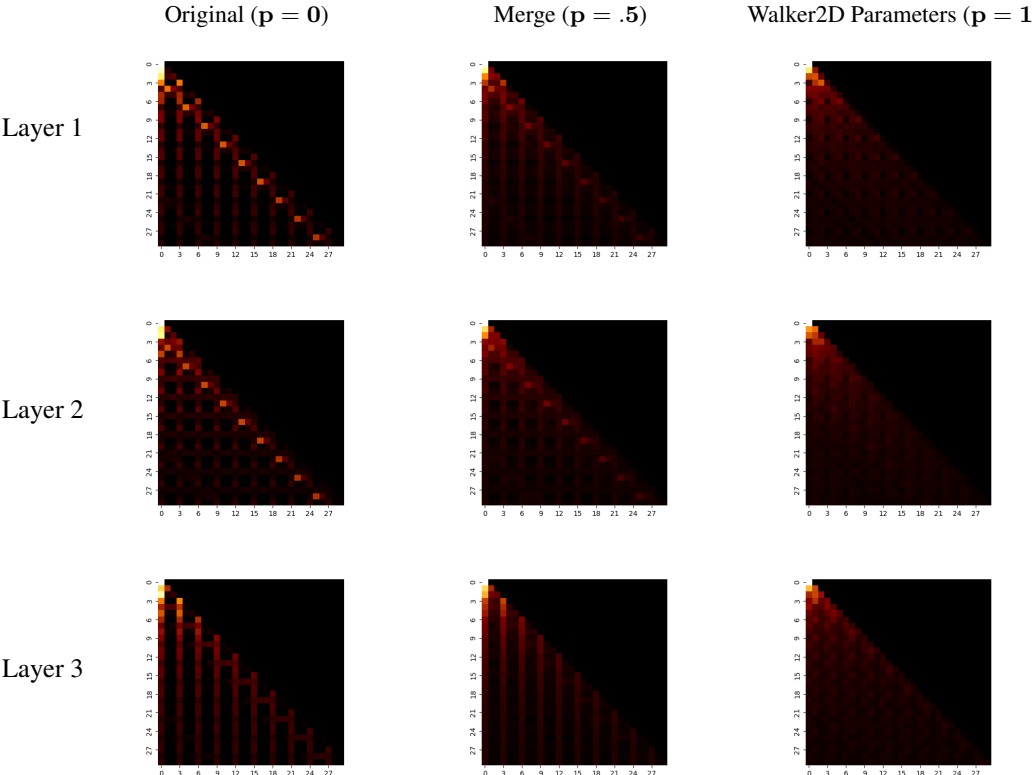

Figure 5: **Attention weight visualizations** on HalfCheetah (Expert), with original parameters ($p = 0$), averaged with Walker2D ($p = .5$), and replacing attention parameters with those from Walker2D ($p = 1$). Under all settings, even as attention weights change, the Decision Transformer retains performance close to that of the original as shown in Figure 4.

separately for each layer, and average weights over a single trajectory sampled from the original training data. We show original attention patterns ($p = 0$), after merging ($p = .5$) and directly using Walker2D parameters ($p = 1$). We can see that at $p = .5$, we have relatively similar attention patterns to $p = 0$, but attention patterns completely change at $p = 1$. This means that the DT can still operate well even when the representations produced by attention change. This could indicate several things. One possibility is that in DTs trained on certain MuJoCo locomotion tasks and dataset combinations, the attention mechanism plays little impact. This could be possible if a model learns to pass more information through residual connections, skipping attention layers. Alternatively, MLP layers following attention could handle variation in representations.

### 3.4.1 PERTURBING ATTENTION PARAMETERS

In this section, we perturb attention parameters after training to see how much randomly initialized DTs rely on the attention mechanism. We replace the attention weights of trained DTs using both random parameters (fixed for all experiments), identity parameters (weights set to 1, and biases equal to 0), and removing attention, having information pass through residual connections. We show the results of these variations in Table 2. We see varied results depending on the environment and dataset pair. On HalfCheetah, we generally see little impact when perturbing attention on all datasets. Given these results, may conclude that attention is not relevant on HalfCheetah. However, Takagi (2022) shows that while HalfCheetah Medium can converge without relying on attention, but it is slower than training with attention. Thus, even when the context may not be needed during inference time, attention over past context can be relevant during training. With Walker2D, we see little impact on homogeneous datasets, Medium and Expert, but see reduced performance from perturbation with Medium-Expert. With Hopper, we see perturbation reduce performance across all settings. This

| Dataset | Environment | Original | Random | Identity | Removed Attention |
|---------|-------------|----------|--------|----------|-------------------|
| Medium-Expert | HalfCheetah | 42.9 | 41.9 | 40.9 | 40.5 |
| | Hopper | 108.3 | 77.2 | 66.8 | 74.1 |
| | Walker2D | 109.4 | 84.6 | 82.6 | 86.9 |
| Medium | HalfCheetah | 40.2 | 39.3 | 40.5 | 39.1 |
| | Hopper | 76.4 | 52.2 | 52.6 | 52.2 |
| | Walker2D | 75.3 | 74.1 | 76.2 | 74.3 |
| Expert | HalfCheetah | 89.8 | 89.2 | 89.2 | 86.2 |
| | Hopper | 108.8 | 67.1 | 67.9 | 68.8 |
| | Walker2D | 109.4 | 109.7 | 109.8 | 109.8 |

Table 2: We report the impact of **altering attention parameters of trained DTs**. We report original performance, replacing attention parameters with randomly initialized parameters, identity parameters, and removing attention, relying on residual connections.

also lines up with Takagi (2022), where training without context in Hopper medium causes training to fail, and not achieve more than 0 mean return.

## 3.5 Merging with Language Pre-training and Co-Training

While we have so far studied merging two randomly initialized DTs as a baseline, we would get much better results if we merge two (or more) models which share a common initialization, as they will likely be much closer in weight space. Additionally, if models are encouraged to stay close to this original initialization, then we may also see better merging results. This could be accomplished through any regularization process, such as adding a shared auxiliary loss, so that we co-train while finetuning. We believe these two strategies are useful for merging DTs, and transformers in general.

We decide to pre-train and co-train with language, as Reid et al. (2022) shows convergence and performance benefits from initializing DTs with ChibiT, a small transformer pre-trained on the Wiketext-103 dataset (Merity et al., 2016). ChibiT has the same architecture and parameter count (596K) as DTs previously used. Training a DT with ChibiT initialization uses the objective $\mathcal{L} = \mathcal{L}_{\text{MSE}} + \lambda_1 \mathcal{L}_{\cos} + \lambda_2 \mathcal{L}_{\text{LM}}$, where $\mathcal{L}_{\text{MSE}}$ is the original DT objective, $\mathcal{L}_{\text{LM}}$ is the language modeling objective, and $\mathcal{L}_{\cos}$ encourages cosine similarity between DT input embeddings and clustered centers of language token embeddings. While $\lambda_1$ is decayed to $0.0$ after 5000 steps in Reid et al. (2022), we maintain $\lambda_2 = 1$ throughout training.

We train models using this procedure for HalfChetah, Hopper, and Walker2D Medium tasks and evaluate both merging, and *Merge-Freeze-Finetune* (MFF) as in Section 3.3, displaying results in Table 3. We merge across all three models, and between pairs, as in previous experiments. Across all pairs, we find we can merge attention layers without any decrease in performance. We find additionally merging MLP layers works well, and with finetuning, we maintain original performance on both HalfCheetah and Walker2D as well as $\sim 94\%$ performance on Hopper. For example, by merging Walker2D and HalfCheetah, we obtain a multi-task model with transformer layers about the size of a single model ($100.26\%$), with no decrease in performance compared to single models, without ever centrally training for both environments. Finally, we see that we can merge all three models at once. This works well with attention merging, and we see significantly reduced performance when adding MLP layers, but we can recover most performance if we also finetune. When applying MFF in this scenario, we obtain an average of $96.27\%$ of original performance across the three environments, with only $1.0052$ times the transformer parameters of a single model (as we retain unique layer normalization layers for each environment, but share MLP/Attention parameters).

## 4 Related Work

Several papers have studied merging models with shared initialization in language and vision for the purpose of improving generalization and task capabilities (Li et al., 2022; Ilharco et al., 2022; Chronopoulou et al., 2023; Jin et al., 2022), increasing robustness to distribution shift (Wortsman

| Method | Models | Hopper | HalfCheetah | Walker | Size |
|---|---|---|---|---|---|
| Attention (M) | Hopper-HalfCheetah | 99.9% | 101.6% | | |
| | HalfCheetah-Walker2D | | 100.3% | 102.0% | 166.82% |
| | Walker2d-Hopper | 101.4% | | 102.0% | |
| | All | 98.9% | 99.4% | 103.1% | 233.64% |
| Attention+MLP (M) | Hopper-HalfCheetah | 54.5% | 83.0% | | |
| | HalfCheetah-Walker2D | | 64.2% | 95.5% | 100.26% |
| | Walker2d-Hopper | 70.5% | | 0.3% | |
| | All | 62.0% | 45.5% | 0.0% | 100.52% |
| Attention+MLP (MFF) | Hopper-HalfCheetah | 94.4% | 101.3% | | |
| | HalfCheetah-Walker2D | | 100.6% | 100.3% | 100.26% |
| | Walker2d-Hopper | 94.9% | | 100.7% | |
| | All | 88.4% | 99.9% | 100.5% | 100.52% |

Table 3: **Merging Decision Transformers trained with language model initialization (ChibiT) and co-training** , on D4RL Medium datasets. We evaluate just merging attention, and attention + MLP layers. We also evaluate *Merge-Freeze-Finetune* (MFF), where we freeze merged layers, and separately finetune the remaining layers. We report merging between both pairs of models and merging all three models at once (All). We report performance relative to before merging which are D4RL normalized scores of 64.5, 40.7, 77.9 for Hopper, HalfCheetah, and Walker2D.

et al., 2021), and for distributed training (McMahan et al., 2016; Don-Yehiya et al., 2022). Other work has improved methods for merging weights by utilizing Fisher information for weighted averaging (Matena & Raffel, 2021), and accounting for symmetries, such as permutation (Ainsworth et al., 2022; Jordan et al., 2022; Peña et al., 2022), to effectively merge weights found through optimizing the same loss but with different initialization. To our knowledge, we are the first work that considers merging for decision-making or robotic control settings, which presents unique challenges. Many recent works have considered creating multi-task or general models for decision-making problems (Lee et al., 2022; Kumar et al., 2022a; Reed et al., 2022; Jiang et al., 2022; Du et al., 2023), but rely on simultaneous and centralized training over all tasks. While we focus on experiments framed as multi-task problems, merging, especially without task-specific finetuning, could be explored for continual (reinforcement) learning (Khetarpal et al., 2020), which has long-standing goals of creating systems that can continuously adapt to new tasks.

## 5 DISCUSSION & CONCLUSION

We believe several extensions and follow-ups would be useful to see if our findings more generally hold. We may expect merging, especially with co-training, to work better on larger models which have more capacity. When merging, we also report results between single task-specific Decision Transformers, and it could be useful to train and report results averaged over several starting models. We also have interests to evaluate merging DTs in other multi-task settings (Lee et al., 2022), testing other kinds of pre-training (Kumar et al., 2022b; Bonatti et al., 2022) besides language, or merging other kinds of models in the RL settings which incorporate pre-training and finetuning (Kumar et al., 2022a; Taiga et al., 2023; Xu et al., 2022).

In this work, we studied merging Decision Transformers trained on different MuJoCo locomotion problems. We found that it is possible to merge randomly initialized models, leading us to analyze the role of attention in DTs. We also investigate merging with additional finetuning, showing we can adapt to changed representations. Lastly, we present a method for obtaining multi-task models utilizing pre-trained language model initializations and co-training. In this setting, we show that we can merge three models, obtaining a multi-task policy without centralized training, and of similar size as a single policy. In general, we view merging as a possible direction for more flexibly creating multi-task DTs or generalist policies.

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

## A  APPENDIX

### A.1  HYPERPARAMETERS & IMPLEMENTATION DETAILS

| Hyperparameters | Value |
|---|---|
| Number of layers | 3 |
| Number of attention heads | 1 |
| Embedding dimension | 128 |
| Batch size | 64 |
| Nonlinearity function | ReLU/GeLU |
| Context length $K$ | 20 |
| Dropout | 0.1 |
| Learning rate | $10^{-4}$ |

Table 4: Hyperparameters used for training Decision Transformers. We use a dropout of 0.1 to match Chen et al. (2021) while Reid et al. (2022) uses 0.2. While ReLU is used when training DTs from scratch, GeLU is used when finetuning from ChibiT, as it uses GeLU activations.

For our experiments, we base our implementation on Reid et al. (2022): `https://github.com/machelreid/can-wikipedia-help-offline-rl`, and provide hyperparameters in Table 4.

### A.2  CROSS-ENVIRONMENT INITIALIZATION

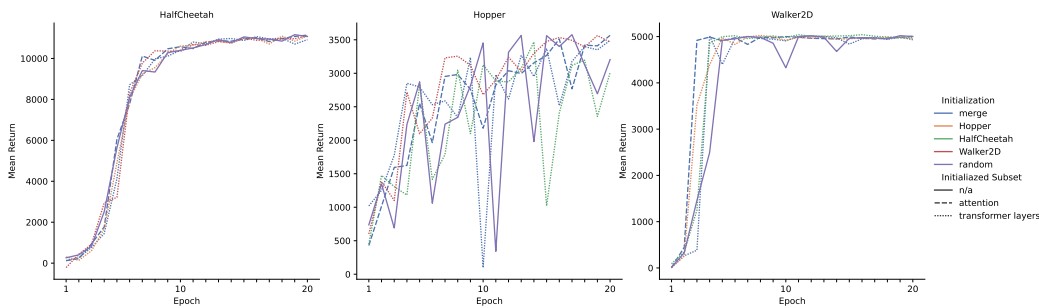

Figure 6: Training with **random initialization versus using initialization from other Decision Transformers** on a different environment, or merged weights from two different DTs. For merged initialization, we evaluate both only initializing attention layers and initializing all transformer layers.

We see if when training a Decision Transformer, we can accelerate training through initialization with another DT trained on another environment, or the merged parameters of multiple DTs. In other work utilizing DTs, (Lawson & Qureshi, 2022) finds convergence benefits when transferring one DT trained for partially-observed maze navigation with a specific robot for initializing training a policy for another robot. We visualize training plots in Figure 6, plotting the mean return after each epoch. We see varied, but not convincing results, where we find improvements on training Walker2D, and minor improvements on HalfCheetah, but little impact on Hopper.

