# OpenReview forum: "Merging Decision Transformers: Weight Averaging for Forming Multi-Task Policies"
_ICLR.cc/2023/Workshop/RRL — RRL 2023 Spotlight_

### Official Review · Reviewer_6TGg · 2023-02-24
**Reusing transformer weights across environments**

**Rating:** 3
**Confidence:** 4

**Review:**

This is very interesting work looking at the performance of a decision transformer (DT), when its weights have been "transplanted" from another DT trained in a different (Mujoco) environment. The authors show that in some circumstances, they can recover performance on original tasks with merged weights, especially with finetuning. They perform ablation studies and analyses to identify which merged components (attention, MLP, Layer Norm) affect the performance the most, and discuss different scenarios. This work fits very well with the workshop goal of reusing prior computation in RL. The paper is well written and easy to read.

Some suggestions/comments/questions:
- Cite prior work in: "Prior work considers merging models trained on the same data distribution, but we begin by using merging as a tool to see the similarity of DTs trained on different data"

- Figure 1: The x-axis can be wrongly interpreted by assuming 17 subsequent layers, where those are actually groupings corresponding to different merging types. It would help if you differently organize/highlight the bars (perhaps space them out differently) and mark which bars correspond to which condition.

- Using Identity Attention Parameters: I found it really interesting that identity parameters gives even higher scores than the Original for Walker2D. Could you speculate why that's the case?

- "For example, we can directly swap in the weights of HalfCheetah into the Walker2D transformer, and vice versa, with no decrease in return" --> is this also true if you just swap in random weights?

- Table 2: Report delta from "Original" instead, or in addition to total scores, and highlight interesting conditions

---

### Official Review · Reviewer_6Y3e · 2023-02-28
**Strong paper with interesting and insightful analysis exploring the effect of merging layers from Decision Transformers trained on independent tasks in order to form a multi-task policy.**

**Rating:** 4
**Confidence:** 4

**Review:**

The paper assesses if it is possible to merge layers from previously trained Decision Transformers (DTs) trained on independent tasks in order to form a multi-task policy that was never co-trained on all the tasks.

The main strengths of the paper are:
* Very relevant to the workshop - the work directly addresses the question of how to re-use previously trained policies.
* Novel and insightful analysis on the effect of merging layers from different DTs on policy performance. This nicely lays the groundwork for future work in the area.
* Well motivated, well written and clearly presented throughout

The main weaknesses are:
* No related work but I think this is fine for a workshop paper.

Overall a strong workshop paper with interesting and insightful analysis that I think form a great basis for discussion in this workshop.